# Structural signatures in EPR3 define a unique class of plant carbohydrate receptors

Jaslyn E. M. M. Wong [1,4], Kira Gysel [1], Thea G. Birkefeldt[1], Maria Vinther[1], Artur Muszyński [2], Parastoo Azadi[2], Nick S. Laursen[1], John T. Sullivan[3], Clive W. Ronson [3], Jens Stougaard [1] & Kasper R. Andersen [1✉]

Receptor-mediated perception of surface-exposed carbohydrates like lipo- and exo-polysaccharides (EPS) is important for non-self recognition and responses to microbial associated molecular patterns in mammals and plants. In legumes, EPS are monitored and can either block or promote symbiosis with rhizobia depending on their molecular composition. To establish a deeper understanding of receptors involved in EPS recognition, we determined the structure of the *Lotus japonicus* (*Lotus*) exopolysaccharide receptor 3 (EPR3) ectodomain. EPR3 forms a compact structure built of three putative carbohydrate-binding modules (M1, M2 and LysM3). M1 and M2 have unique βαββ and βαβ folds that have not previously been observed in carbohydrate binding proteins, while LysM3 has a canonical βααβ fold. We demonstrate that this configuration is a structural signature for a ubiquitous class of receptors in the plant kingdom. We show that EPR3 is promiscuous, suggesting that plants can monitor complex microbial communities though this class of receptors.

[1] Department of Molecular Biology and Genetics, Aarhus University, 8000 Aarhus C, Denmark. [2] Complex Carbohydrate Research Center, University of Georgia, Athens, GA 30602, USA. [3] Department of Microbiology and Immunology, University of Otago, Dunedin 9054, New Zealand. [4] Present address: MRC Laboratory of Molecular Biology, Cambridge CB2 0QH, UK. ✉email: kra@mbg.au.dk

Nitrogen-fixing symbiosis between legumes and rhizobia is governed by a two-step receptor-mediated recognition mechanism[1]. In the first step, rhizobial lipochitooligosaccharides (LCOs or Nod factors) are perceived by plant LCO receptors, which induces the development of root nodule primordia, the entrapment of rhizobia in root hair curls and triggers the expression of symbiotic genes for bacterial infection, including *Epr3*[1–5]. The second step controls the subsequent progression of nodule infection and is mediated by the single-pass transmembrane receptor kinase EPR3[1,5–9]. The natural symbiont of *Lotus*, *Mesorhizobium loti* (R7A), synthesises EPS polymers built from octasaccharide-repeating units. Monomeric octasaccharides (R7A EPS)[10] are recognised through direct binding to EPR3[1]. Studies of rhizobia and host plant mutants show that EPS perception and subsequent EPR3 signalling promotes infection of the epidermal and cortical tissues of *Lotus* and *Medicago* roots[1,5,11]. In contrast, truncated EPS produced by the *exoU* mutant strain (R7A exoU EPS) blocks rhizobial infection and colonisation in an EPR3-dependent manner, suggesting that the perception of EPS is an additional compatibility-determining step in legume–rhizobia interactions[1,6]. Here, we determined the structure of the defining member of a conserved and unique class of plant EPS receptors and show that EPR3 is capable of directly perceiving EPS from different bacterial species, suggesting a broader role in surveillance of microbial communities.

## Results and discussion

**The crystal structure of EPR3**. To understand the basis of EPS perception, the ectodomain of *Lotus* EPR3 (hereafter referred to as EPR3) was expressed in insect cells and purified for structural studies. Despite numerous attempts, EPR3 did not crystallise. We therefore generated a miniature llama-derived antibody (nanobody) to facilitate crystallisation. We raised an immune response against EPR3 by immunising a llama and selected nanobodies by phage display[12,13]. The high-affinity nanobody, Nb186, forms a stable complex with EPR3 as demonstrated by a mobility shift in size-exclusion chromatography (SEC) experiments (Supplementary Fig. 1a). The co-purified deglycosylated-EPR3-Nb186 complex was isolated (Supplementary Fig. 1b, c) and crystallised, and the structure was determined and refined to 1.9 Å resolution (Supplementary Fig. 2 and Supplementary Table 1). The overall structure of EPR3 consists of three interconnected modules (M1, M2 and LysM3) arranged in a cloverleaf-shape and stabilised by three internal disulfide bridges (Fig. 1a). The crystal structure of EPR3 reveals an M1 fold that is structurally unique (Fig. 1b). M1

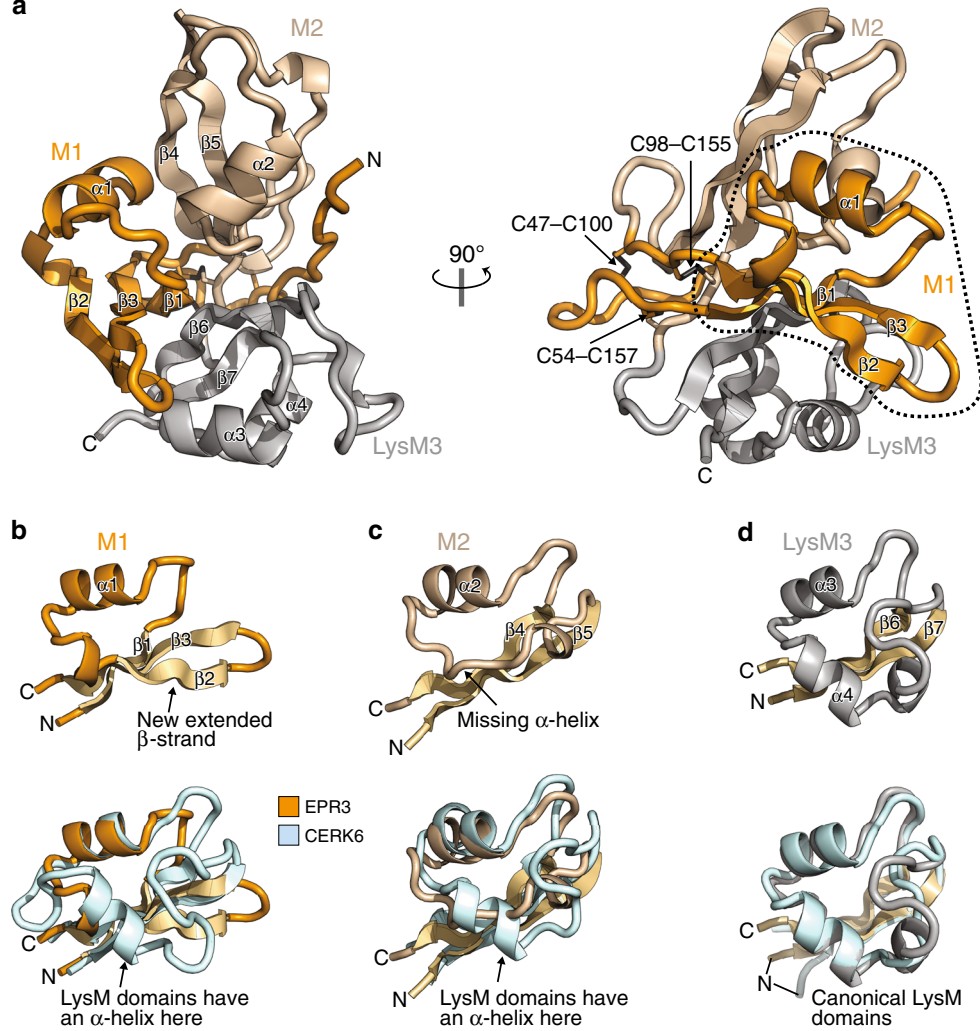

**Fig. 1 The crystal structure of EPR3. a** Cartoon representation of the EPR3 crystal structure with each of the modules M1, M2 and LysM3 coloured in orange, brown and grey, respectively. Secondary structure elements and disulfide bridges are indicated. The dotted line highlights the unique M1 domain. **b–d** Individual carbohydrate-binding modules M1, M2 and LysM3 of EPR3 with labels indicating their secondary structures. The panels below show the modules superimposed on the corresponding LysM domains of CERK6 (PDB - 5LS2) coloured in light blue.

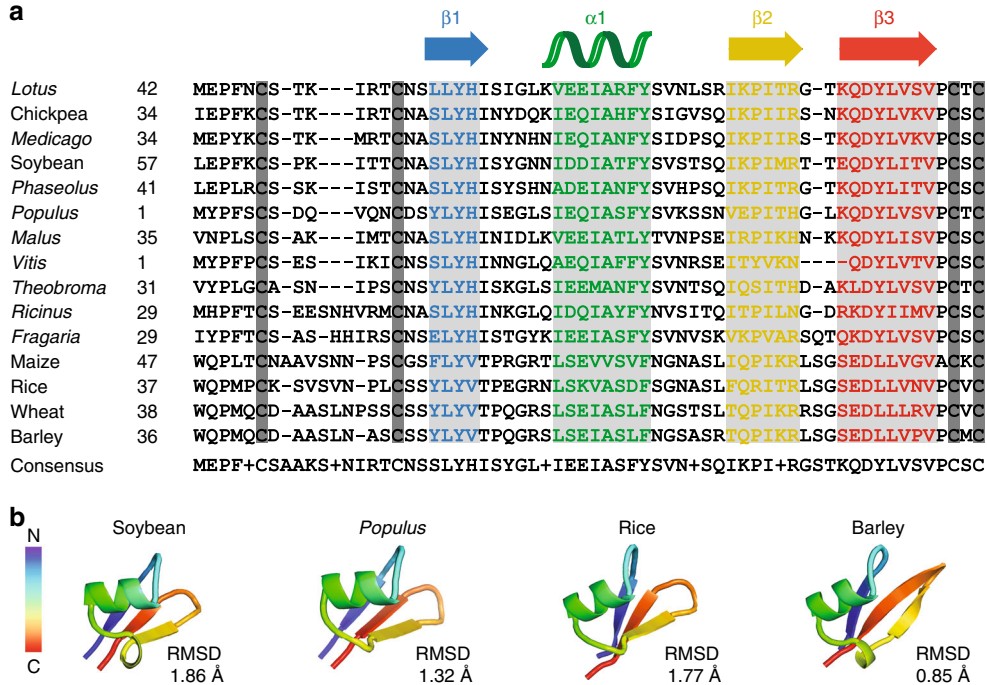

**Fig. 2 M1 is a defining feature of a unique class of plant receptors. a** Amino acid sequence alignment of M1 from *Lotus* EPR3 and EPR3 receptor homologues found in dicots (legumes and non-legumes) and monocots showing the conserved βαββ secondary structure arrangement. **b** Ab initio models of the EPR3-M1 domain from receptor homologues reveal conserved βαββ structures. Molecular fits (RMSD values) based on superposition of these modelled M1 domains to the M1 domain in the EPR3 crystal structure are denoted in Å (Angstrom).

is composed of only one α-helix and three elongated β-strands. The exterior β2-strand is stabilised by seven backbone hydrogen bonds to the adjacent β3-strand, which gives M1 an overall βαββ arrangement where the three β-strands form an extended antiparallel β-sheet (Fig. 1b). The M2 domain of EPR3 is also unusual, as it contains a βαβ fold and lacks the defined second α-helix compared with a canonical LysM domain (Fig. 1c). LysM3 has the standard βααβ fold of LysM proteins, with a root-mean-square deviation (RMSD) of 1.2 Å to the LysM3 domain of *Lotus* chitin receptor CERK6[14] (Fig. 1d). A DALI search in the Protein Data Bank (PDB) revealed that M1 in EPR3 has no close structural homologues and therefore constitutes a unique fold in carbohydrate-binding proteins, while M2 is associated with LysM structures and LysM3 classifies as a standard LysM motif[15,16].

**M1 is a defining feature of a unique class of plant receptors.** The primary sequence and secondary structure of EPR3 with unique N-terminal M1 (βαββ) and atypical M2 (βαβ) folds, followed by a classical LysM3 domain (βααβ) is highly conserved across plant species and defines a unique class of receptors (Fig. 2a; Supplementary Fig. 3). This class of receptors is not restricted to legumes but is also present in non-legume dicots and monocot plants, suggesting that surveillance of EPS or other microbial surface carbohydrates is a widely conserved plant trait. Modelling of this small M1 domain (~43 amino acids) in EPR3 homologues using atomic-level force field simulations reinforces this observation of a structurally conserved class of EPR3 receptors. We find that all de novo built models of EPR3 homologues from 14 different species share the same topology, βαββ fold and superpose well with the crystal structure of *Lotus* EPR3-M1 domain (Fig. 2b; Supplementary Fig. 4). M1 of these receptors forms a surface-exposed β-sheet structurally different from all known carbohydrate-binding modules identified in nature so far[17]. Together, this demonstrates that the M1–M2–LysM3

configuration of EPR3 is a signature of a ubiquitous conserved class of plant receptors that are evolutionarily and structurally distinct from the chitin LysM receptors (Supplementary Fig. 5)[14,18,19]. Although this class of receptors is widespread in the plant kingdom, none, apart from *Lotus* and *Medicago* EPR3, has so far been functionally characterised in planta, which opens a broader line of receptor research.

**EPR3 is monomeric and contains a stem-like structure.** We further performed small-angle X-ray scattering (SAXS) experiments to explore the structure of EPR3 in solution (Fig. 3; Supplementary Table 2). To our surprise, the distance-distribution plot shows that the length of EPR3 is almost twice in solution compared with the crystal structure, despite maintaining the same molecular weight corresponding to a monomer (Fig. 3c; Supplementary Table 2). The low-resolution ab initio SAXS envelope reveals a globular shape with a protruding stem-like structure (Fig. 3d). The crystal structure of EPR3 is well accommodated in the globular part of the SAXS envelope, but leaves the protruding density unaccounted for. Modelling the C-terminal residues, missing in the electron density of the crystal structure, as a stem-like structure improved the fit to the measured SAXS scattering curve (Fig. 3a, d). The stem region shows conservation among EPR3 homologues both in terms of length and composition, which is dominated by glycine and positively charged lysine and arginine residues (Fig. 3e). We hypothesise that the stem serves as a spacer to the plasma membrane with potential importance for efficient signalling or interaction with possible co-receptors (Fig. 3f). However, the functional role of this stem-like structure remains to be validated in future studies.

**EPR3 is a promiscuous EPS receptor.** Perception of compatible EPS in legumes is thought to promote infection of bacteria and to deny entry of incompatible strains to the root[1,5–8]. To understand

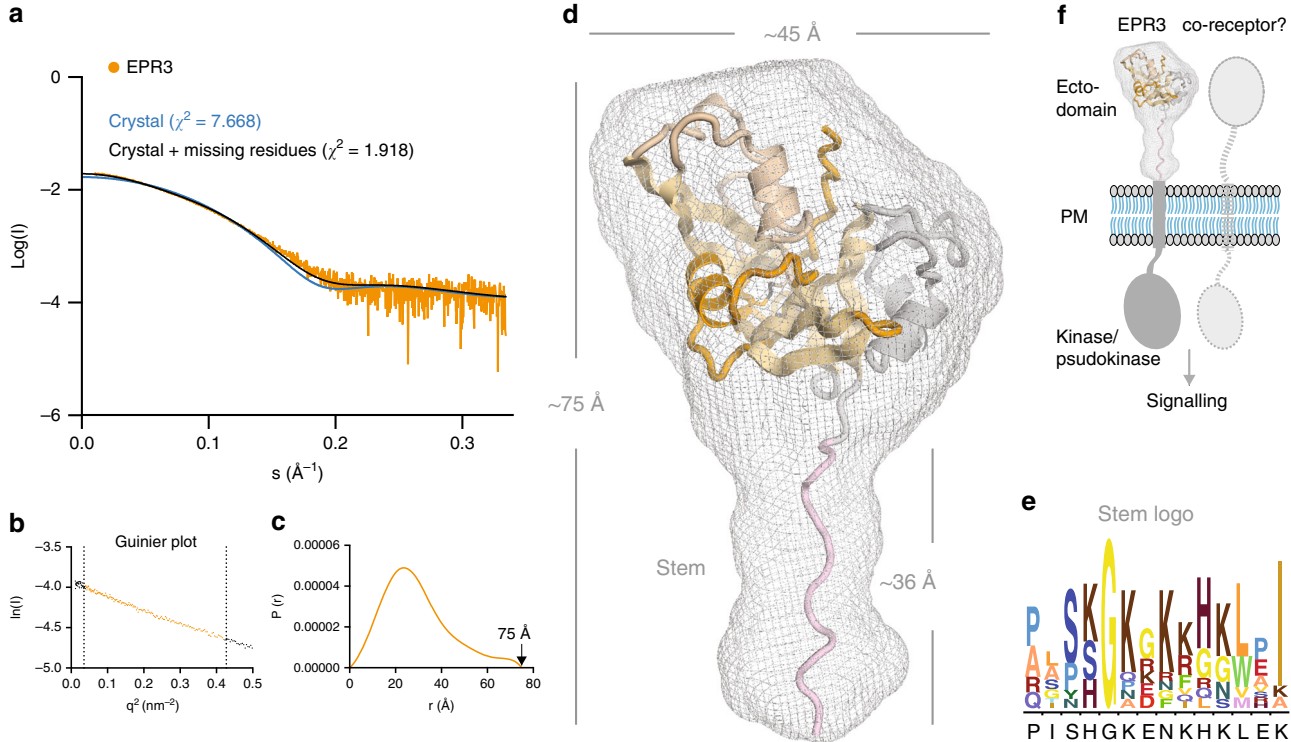

**Fig. 3 EPR3 is monomeric and contains a stem-like structure. a** SAXS analysis of EPR3 showing scattering curve with fit to the crystal structure alone ($\chi^2 = 7.668$) and fit to the crystal structure rebuilt with the missing residues ($\chi^2 = 1.918$). **b** Guinier plots and **c** P(r) distance-distribution plot with $D_{max} = 75$ Å. **d** Rigid-body modelling of the EPR3 crystal structure into the SAXS envelope shows an extended stem-like structure. The overall dimensions are shown in angstrom (Å). **e** Alignment logo of the stem region of EPR3 homologues with the sequence of *Lotus* EPR3 shown below[43]. **f** Model of the EPR3 receptor where the stem structure positions the ectodomain with a distance to the plasma membrane (PM) possible for efficient signalling or interaction to a co-receptor.

the ability of EPR3 to distinguish between EPS of different structures and compositions, we examined ligand binding in solution using microscale thermophoresis (MST) (see all ligands in Supplementary Fig. 6). EPR3 binds the compatible monomeric octasaccharide R7A EPS with an equilibrium-dissociation constant ($K_d$) of $38.1 \pm 7.5\,\mu$M (Fig. 4a). The binding is neither affected by the glycosylation state of the receptor nor the binding of Nb186 (Supplementary Fig. 7), indicating that the EPR3-Nb186 crystal structure is most likely presented in its biologically active state. To detect if ligand binding affects EPR3 oligomerization as a potential signalling mechanism, we determined the SAXS solution structure of EPR3 saturated with R7A EPS. The scattering data and ab initio reconstructions show that the ligand-bound receptor retains its monomeric state with the same overall structure, dimensions and stem-like arrangement as the ligand-free state (Supplementary Fig. 8; Table 2). This could mean that EPR3 signals as a monomer, or, maybe more likely, associates with as yet unidentified co-receptor(s) to form a ligand-induced signalling complex as known from other single-pass transmembrane receptor kinases (Fig. 3f)[2]. To further investigate ligand selectivity, we first assessed if EPR3 binds the immune response-inducing chitin polymer (CO6) known to be perceived by canonical LysM receptors[14,18,19] and found that EPR3 is unable to bind CO6 (Fig. 4b). Next, we examined the ability of EPR3 to recognise symbiotic signalling LCO molecules using the Bio-layer interferometry methodology previously used to show robust *M. loti* Nod factor binding to *Lotus* Nod factor receptors NFR1 and NFRe[20]. EPR3 binding to LCO was not detected. Altogether, our biochemical analyses support that EPR3 indeed belongs to a functional and unique class of receptors, as shown by the

structure. The *N*-acetyl groups of chitin polymers have previously been demonstrated to be important contact points for LysM proteins[18,21–23]. Therefore, we investigated if the corresponding *O*-acetyl groups in EPS are important moieties recognised by EPR3. However, chemical removal of the *O*-acetyl groups in R7A EPS (deOAc-EPS) did not affect binding ($K_d = 31.3 \pm 11.7\,\mu$M, which is similar to that of fully *O*-acetylated R7A EPS) (Fig. 4c). This implies a difference in the ligand perception mechanism between EPR3 and LysM receptors binding chitinous ligands, e.g. *At*CERK1[18]. In the crystal structure of *At*CERK1, the position of chitin in the LysM2-binding pocket allows the carbonyl oxygen of the *N*-acetyl moieties to form hydrogen bonds with backbone amide nitrogens of the main chain[18]. Such rigorous recognition is unlikely for the *O*-acetyl groups in EPS as EPS are non-stoichiometrically *O*-acetylated, in contrast to chitin that has a uniform distribution of *N*-acetyl groups[10]. Supporting this notion, we purified and characterised EPS from both *R. leguminosarum* and *S. meliloti* with different *O*-acetylation patterns (Supplementary Fig. 6) and found that these EPS can still be perceived by EPR3. The production of diffusible octasaccharide monomers, corresponding to the main EPS polymer subunits, is not exclusive to R7A, but is present also in *R. leguminosarum* and *S. meliloti*, and likely also in other rhizobia, suggesting an important role. Although these rhizobia do not normally infect *Lotus*, their secreted EPS are detected by EPR3. EPR3 binds *R. leguminosarum* octasaccharide EPS with a $K_d = 9.0 \pm 3.7\,\mu$M (Fig. 4d) and *S. meliloti* EPS (succinoglycan) with a $K_d = 221.9 \pm 102.3\,\mu$M (Fig. 4e, f), demonstrating that *Lotus* EPR3 is a promiscuous receptor capable of surveying EPS from different bacterial species while selectively discriminating against

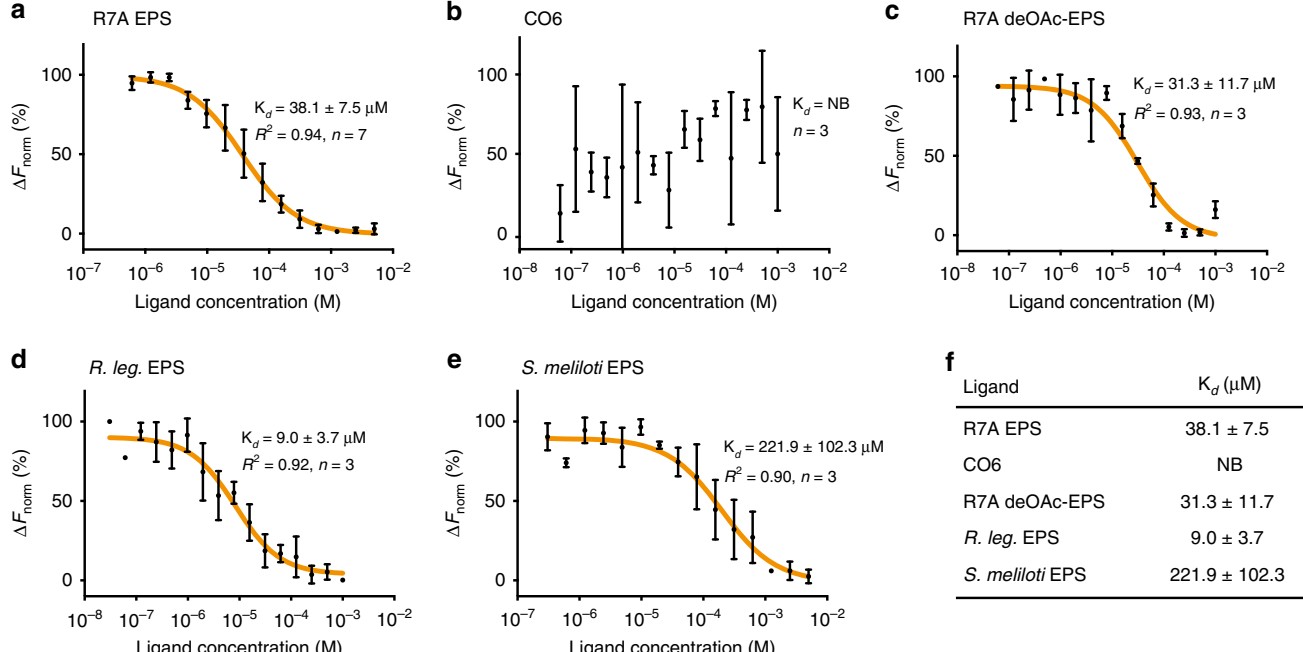

**Fig. 4 Lotus EPR3 is a promiscuous EPS receptor. a** EPR3 binds R7A EPS with a $K_d$ of 38.1 ± 7.5 μM ($n = 7$). **b** EPR3 does not bind chitin (CO6) ($n = 3$). **c** EPR3 binds R7A de-O-acetylated EPS (deOAc-EPS) with a $K_d$ of 31.3 ± 11.7 μM ($n = 3$). **d** EPR3 binds *R. leguminosarum* (*R. leg.*) EPS with a $K_d$ of 9.0 ± 3.7 μM ($n = 3$) and **e** *S. meliloti* EPS with a $K_d$ of 221.9 ± 102.3 μM ($n = 3$). **f** Overview of the equilibrium-dissociation constants value ($K_d$) in the 95% confidence interval for the different ligands. The corresponding goodness of fit ($R^2$) are indicated, and ($n$) represents the number of replicates performed using independent protein preparations. NB no detectable binding.

carbohydrates, such as chitin and maltohexose[1]. One explanation for the stronger EPR3 binding of *R. leguminosarum* octasaccharide is that *R. leguminosarum* EPS may be perceived as compatible by *Lotus*. This explanation is supported by the earlier observation that an *R. leguminosarum* DZL strain engineered to produce a Nod factor, similar to the Nod factor produced by the *Lotus M. loti* symbiont, can form infected and nitrogen-fixing nodules, albeit with a delay[24,25]. *R. leguminosarum–Lotus* incompatibility appears to be governed by Nod factor recognition, the first of the two-step recognition mechanism, and not by the second step of EPS recognition. Taken together with the widespread conservation of the EPR3 class of receptors among plants, these results imply that EPR3 and the homologs in non-legume plants are most likely surveillance receptors monitoring carbohydrates from different microbes associated with plant roots. Bacterial EPS is only one group of such carbohydrates. Another example is short-chain and long-chain beta-glucans of bacterial or fungal origin, some of which have been shown to elicit responses in many plants[26,27].

In summary, we demonstrate that EPR3 is a defining member of a large and conserved unique class of plant receptors capable of directly perceiving EPS from different bacterial species. This evolutionary conservation highlights a widespread requirement for plants to recognise EPS or other microbial surface carbohydrates, possibly for monitoring associated microbiota. EPR3 contains an intracellular kinase domain predicted to be active. Based on our current knowledge of LysM receptor systems[2], it is likely that EPR3 is assisted by a co-receptor containing an inactive pseudokinase domain (Fig. 3f). Future studies in different plant species will help us to better understand this class of receptor and its downstream signalling mechanisms.

## Methods

**Protein production.** Expression and purification of *Lotus japonicus* ecotype Gifu EPR3 was performed as described previously[1]. In brief, DNA encoding residues

33–232 of EPR3 containing an N-terminal gp67 secretion signal and a C-terminal 6xHis-tag was codon-optimised for insect cell expression (GenScript) and inserted into the pOET2 vector (Oxford Expression Technologies). Recombinant baculovirus, used for infecting Sf9 cells cultured in suspension in serum-free HyClone SFX-Insect medium (Fisher Scientific), was obtained using the flashBAC GOLD system (OET). Five days post infection, the media was dialysed into buffer containing 50 mM Tris-HCl pH 8.0 and 200 mM NaCl before centrifugation and loaded on a HisTrap excel affinity column (GE Healthcare). The eluted protein was dialysed in buffer containing 50 mM Tris-HCl pH 8.0 and 200 mM NaCl, and further purified on a second HisTrap HP affinity column (GE Healthcare). For crystallisation, EPR3 was treated with PNGase F (1:15 w/w ratio) for 1 h at room temperature and overnight at 4 °C to remove N-linked oligosaccharides. EPR3 was then purified on a Mono S 5/50 column (GE Healthcare) and eluted with a linear gradient of 50–300 mM NaCl and 50 mM Tris-HCl, pH 7.0. Both glycosylated and de-glycosylated EPR3 were purified on a Superdex 75 10/300 column (GE Healthcare) in buffer containing 50 mM KH$_2$-PO$_4$ pH 7.8 and 200 mM NaCl (for MST-binding experiments) or 50 mM Tris-HCl pH 8.0 and 200 mM NaCl (for crystallisation and SAXS).

**Nanobody production.** A llama (*Lama glama*) was immunised four times with 100 μg of purified EPR3. Peripheral blood lymphocytes were isolated from a blood sample, and RNA was extracted using RNase Plus Mini Kit (Qiagen). The total cDNA was generated using the Superscript III First-Strand Kit (Invitrogen) with random hexamer primers. The coding regions of the nanobodies (Nbs) were amplified by PCR, and inserted into a phagemid vector backbone where the Nbs were C-terminally fused to an E-tag followed by the pIII coat protein. VCSM13 helper phage was used for generating the final M13 phage display Nb library. For selection, EPR3 was biotinylated via primary amine coupling using the Chromalink NHS labelling system (Solulink) and 20 μg of EPR3 antigen was added to 100 μl of MyOne Streptavidin T1 Dynabeads (Thermo Fisher Scientific) in PBS supplemented with 2% BSA. M13 phage particles (2.5 × 10¹³) were added and incubated with EPR3-coated Dynabeads for 1 h before 15 wash steps with 1 ml of PBS containing 0.1% Tween-20. Phages were eluted by incubating the beads with 0.2 M glycine pH 2.2 for 15 min. The eluted phage particles were amplified and used in a second round of phage display where a reduced amount of EPR3 antigen (2 μg) and fewer M13 phage particles (2.5 × 10¹²) were used. After two rounds of phage display selections, single colonies were picked and grown in LB medium in a 96-well plate format for 6 h before Nb expression was induced with 0.8 mM IPTG overnight at 30 °C. The 96-well plate was centrifuged, and 50 μl of the supernatant were transferred to an EPR3-coated ELISA plate prepared by coating each well with 0.1 μg of EPR3 and by blocking with PBS containing 0.1% Tween-20 and 2% BSA. After addition of the supernatant, the EPR3-coated ELISA plate was incubated for 1 h, and then washed six times in PBS with 0.1% Tween-20 before anti-E-tag-HPR

antibody (Bethyl) was added at a 1:10.000 dilution. The plate was incubated for 1 h, washed and developed with 3,3',5,5'-tetramethylbenzidine. The reaction was quenched with 1 M HCl, and the absorbance was measured at 450 nm. Phagemids from positive clones were isolated, sequenced and the encoding DNAs were cloned into the pET22b(+) (Novagen) for bacterial expression. Nb186 was expressed in *E. coli* LOBSTR cells[28] that were grown to an optical density of 0.6 at 600 nm before protein expression was induced with 0.2 mM IPTG at 18 °C overnight. Cells were lysed in buffer containing 50 mM Tris-HCl pH 8.0, 500 mM NaCl, 20 mM imidazole and 1 mM benzamidine, and the cleared supernatant was loaded onto a Ni Sepharose 6 FF affinity column (GE Healthcare) and washed prior to elution in lysis buffer supplemented with 500 mM imidazole. Nb186 was finally purified on a Superdex 75 10/300 gel filtration column (GE Healthcare) in buffer containing 50 mM Tris-HCl pH 8.0 and 200 mM NaCl. Complex formation between EPR3 and Nb186 was analysed on an analytic Superdex 75 Increase 3.2/300 column (Supplementary Fig. 1a).

**Crystallisation and structure determination.** Purified de-glycosylated EPR3 and Nb186 were mixed in a 1:1.1 molar ratio and incubated on ice for 1 h before purification on a Superdex 75 10/300 column. The peak fractions containing the EPR3-Nb186 complex were pooled and concentrated on a VivaSpin filter (Sartorius) to 5–8 mg/ml and crystallised using the vapour-diffusion method by mixing an equal volume of protein and reservoir solution (18% 2-propanol (v/v), 0.1 M sodium citrate pH 5.5 and 20% PEG 4000 (v/v)). Crystals were cryoprotected in mother liquor with the addition of 20% ethylene glycol before being flash-frozen in liquid nitrogen. Diffraction data were measured at DESY P14 beamline at a wavelength of 0.9763 Å, and data reduction was performed in XDS[29]. A molecular replacement solution was found with phenix.phaser[30] using a homology model of Nb186 generated with Phyre2[31] truncated of its complementarity-determining regions (CDRs). In a second molecular replacement search, a homology model of EPR3 generated with Phyre2 and truncated of high b-factor region based on CERK1 structure (PDB entry 4EBZ) was placed. The structure of the EPR3-Nb186 complex was built in Coot[32] and coordinates and temperature factors were refined using phenix.refine[33]. The final model contained residues 1–119 of Nb186 and residues 36–216 of EPR3 with 98% of the protein residues in the favoured region and none in the disallowed region of the Ramachandran plot. The figures were prepared with PyMOL, and data and refinement statistics are summarised in Supplementary Table 1.

**Modelling.** De novo modelling of the M1 domain of *Lotus* EPR3 and EPR3 homologues (corresponding to residues 56–99 in EPR3) was performed using atomic-level knowledge-based force field simulations[34].

**Characterisation of EPS ligands.** Low-molecular mass (LMM) exopolysaccharides (EPS) were isolated from rhizobial strains *M. loti* R7A ndvB[6], *R. legumniosarum* bv. *viciae* 3855[35] and *S. meliloti* B578[36] that were deficient in cyclic glucan production. All strains were grown in minimal medium with glucose as the sole source of carbon. The LMM EPS was isolated from the bacterial culture supernatants and purified via sequential precipitation with three volumes of 99.8% EtOH (v/v), followed by nine volumes EtOH (v/v) and purified by SEC, as previously described[10]. O-acetyl groups were removed chemically by mild overnight treatment of native EPS samples with 12.5% NH$_4$OH[10]. Native and de-O-acetylated samples were verified via MALDI-TOF-MS analysis on Applied Biosystems AB SCIEX TOF/TOF 5800 system in either negative or positive reflector ionisation modes. The glycosyl composition and linkage was determined as previously described[10].

**Native and de-O-acetylated R7A EPS.** We previously demonstrated that *M. loti* R7A produces a LMM EPS that is structurally similar to HMM EPS polymer, and is an O-acetylated octasaccharide with the structure (2,3/3OAc)β-D-Rib*f*A-(1 → 4)-α-D-Glc*p*A-(1 → 4)-β-D-Glc*p*-(1 → 6)-(3OAc)β-D-Glc*p*-(1 → 6)-(2OAc)β-D-Glc*p*-(1 → 4)-(2/3OAc)β-D-Glc*p*-(1 → 4)-β-D-Glc*p*-(1 → 3)-β-D-Gal*p*, and the average molecule is substituted with three O-acetyl groups at four glycosyl residues in a non-stoichiometric ratio[10]. In this work, we repeated the experiment and found similar structural properties of the isolated LMM EPS. In particular, MALDI-TOF-MS analysis confirmed that the average molecular [M–H]⁻ mass of the R7AΔndvB EPS was m/z 1437.40, consistent with RibAGlcAGlc$_5$GalOAc$_3$ (Supplementary Fig. 6a). De-O-acetylation of the wild-type native R7AΔndvB EPS resulted in a shift of molecular mass from m/z 1437.40 to m/z 1311.18. This is consistent with loss of all three O-acetyl groups from RibAGlcAGlc$_5$Gal octasaccharide (Supplementary Fig. 6b).

**Chitohexaose (CO6).** CO6 was obtained from Megazyme (Supplementary Fig. 6c).

**R. leguminosarum EPS.** An ndvB mutant of *R. leguminosarum* bv *viciae* 3855 was constructed by insertion of a suicide vector into the ndvB gene, as previously described[1,6]. Native *R. leguminosarum* 3855 ndvB EPS SEC purification of nine volumes EtOH precipitated EPS yielded one major low-molecular mass fraction (LMM EPS). *R. leguminosarum* bv. *viciae* 3855, produces an octasaccharide EPS polymer consisting of five β-D-Glc*p*, two β-D-Glc*p*A, and one β-D-Gal*p* residues substituted with three 2-O-acetyl (or 3-O-acetyl), two 4,6-pyruvyl and one hydroxybutanoyl (OHB) group[37–39]. Composition and glycosyl linkage analysis indicated the presence of 4-substituted Glc*p*, 6-substituted Glc*p*, 4-substituted Glc*p*A, 4,6-disubstituted Glc*p*, 4,6-disubstituted Gal*p* 3,4,6-trisubstituted Glc*p* (all branching sugars likely due to 4,6-substitution with pyruvate), and terminal Glc*p*. Negative ionisation mode MALDI-TOF-MS analysis demonstrated a heterogeneous mixture of Hex₆HexA₂ octasaccharide with a different number of non-carbohydrate substituents, and major [M–H]⁻ ion at m/z 1565.37, likely due to the fact that octasaccharide was substituted with two O-acetyl and two 4,6-pyruvyl groups. We also detected the structures substituted with hydroxybutanoate, but these are not major moieties (Supplementary Fig. 6d).

**S. meliloti EPS.** SEC purification of precipitated *S. meliloti* EPS yielded one major low-molecular mass fraction (LMM EPS). *S. meliloti* B587 is an ndvB mutant of Rm1021 that is proposed to be deficient in cyclic glucan production while producing normal EPS[36]. The Rm1021 EPS (succinoglycan or EPS-I) is an octasaccharide polymer consisting of seven β-D-Glc*p* and one β-D-Gal*p* residues substituted with 6-O-succinyl, 6-O-acetyl, and 4,6-pyruvyl groups. Composition and glycosyl linkage analysis indicated the presence of 3-substituted Gal*p*, 4-substituted Glc*p*, 6-substituted Glc*p*, 3-substituted Glc*p*, 4,6-disubstituted Glc*p* (likely due to 4,6-substitution with pyruvate). Consistent with early reports[36], no 2-substituted glucose was detected, confirming there was no cyclic glucan production. Negative ionisation mode MALDI-TOF-MS analysis indicated major [M–H]⁻ ion at m/z 1525.20. This ion corresponds to an octasaccharide composed of eight hexose residues substituted with O-acetyl, 4,6-pyruvyl and succinyl groups (Hex8OAcOSucPyr) (Supplementary Fig. 6e).

**MST-binding experiments.** Purified EPR3 was fluorescently labelled using the Monolith NT.115TM Protein Labelling Kit Blue NHS (NanoTemper Technologies) according to the manufacturer's instructions. All experiments were performed in MST buffer (50 mM K₂PO₄, pH 7.8, 500 mM NaCl and 0.05% Tween-20) with a constant concentration of EPR3 (100 nM and ~50% labelling efficiency) and dilution series of the various ligands. The samples were incubated for 30 min at room temperature before being loaded into standard capillaries for measurements on a Monolith NT.115 TM instrument (NanoTemper Technologies) at 25 °C, with blue LED power of 50% and MST power of 20%. To accurately measure the experimental errors and ensure data reproducibility, all MST-binding experiments were performed with at least three independently purified samples of EPR3. At the highest ligand concentrations, we occasionally observed weak ligand binding to the fluorescent label itself. To accurately account for this unspecific binding we measured ligand binding to 50 nM free fluorescent label and subtracted this small background contribution to all the respective MST-binding measurements. Binding data were processed with the Prism 7 software (GraphPad Software, Inc.), and the equilibrium-dissociation constant ($K_d$) values (95% confidence interval) were calculated using the sigmoidal dose–response model.

**SAXS.** A monodisperse peak fraction of EPR3 was collected from a SEC experiment and used for SAXS measurements. Scattering from EPR3 samples without ligand or with R7A EPS (1 mM) added at concentrations ranging from 0.6 to 22.0 mg/ml were collected at the EMBL PETRA III P12 beamline in a temperature-controlled cell (20 °C) at a wavelength of 1.24 Å. Normalisation and radial averaging were done at the beamline using the automated pipeline. Buffer subtraction and further data analysis were done in primusqt and BioXtas RAW[40]. Ab initio low-resolution modelling was performed in DAMMIF (15 modelling runs) before averaging in DAMAVER and a final refinement step in DAMMIN[41]. The theoretical scattering profiles of the atomic structures and the experimental data fit were calculated using CRYSOL[42]. The scattering, Guinier plots and P(r) distance-distribution plots were prepared with the Prism 7 software (GraphPad Software, Inc.).

**Reporting summary.** Further information on research design is available in the Nature Research Reporting Summary linked to this article.

## Data availability
The coordinates and structural factors for the crystal structure of EPR3 has been deposited in the Protein Data Bank under PDB code 6QUP. Source data are provided with this paper.

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

## Acknowledgements

We thank Arshia Ghodrati and Joshua B. Nunez for their help in purifying EPS from *R. leguminosarum* and *S. meliloti* and Mickaël Blaise, Simon Hansen, Simon Kelly and Simona Radutoiu for valuable insights and discussions. The work was supported by the Danish National Research Foundation (DNRF79) and by the project Engineering Nitrogen Symbiosis for Africa (ENSA; OPP11772165) currently supported through a grant to the University of Cambridge by the Bill & Melinda Gates Foundation and UK government's Department for International Development (DFID). Work at Center for Plant and Microbial Complex Carbohydrates at the Complex Carbohydrate Research Center was supported by the U.S. Department of Energy (DOE), Office of Science, Basic Energy Sciences (BES) (DE-SC0015662).

## Author contributions

J.E.M.M.W.: crystal structure, biochemistry and study design; K.G.: SAXS, biochemistry and study design; T.G.B.: biochemistry; M.V.: protein production; A.M. and P.A.: ligand purification and characterisation; N.S.L.: nanobody production; J.T.S., C.W.R. and A.M.: ligand production; J.S.: study design; K.R.A.: crystal structure, nanobody production and study design. K.R.A. wrote the paper with input from all authors.

## Competing interests

J.E.M.M.W., K.G., J.S and K.R.A. are inventors on a patent application (62888944) submitted by Aarhus University entitled: Modified exopolysaccharide receptors for recognising and structuring microbiota. The remaining authors declare no competing interests.
