## [Peer Review File · Nature Communications]

REVIEWER COMMENTS

Reviewer #1 (Remarks to the Author):

Review: Structural signatures in EPR3 define a new class of plant carbohydrate receptors.

In their manuscript, Wong et al., determine and describe the apo structure of the *Lotus japonicus* EPR3 receptor ectodomain. The structure reveals a unique domain arrangement to sense carbohydrate ligands. The authors also explore some structural features of the receptor-ECD in solution using SAXs. SAXs data reveal an interesting protruding stem-like structure (not visible in the crystal structure) with unknown function. Using SAXs analysis of the receptor, in the presence or absence of the EPS ligand, they hypothesize that the receptor-ligand complex may require a potential co-receptor for activation. The authors also define the binding affinities of different EPS ligands to the receptor, showing its capacity to recognize different EPS ligands from different bacterial species, however, being able to distinguish other defense signaling sugars such as chitin. The authors also test in vitro the potential relevance of the O-acetyl groups for the recognition of the sugar ligand by the receptor.

The paper is nicely written and easy to follow, and the receptor ECD apo structure is indeed a novel element in the manuscript to be acknowledged and will provide a great platform to move forward in the understanding on how this signaling mechanism works; however, the authors do not exploit the structural data results nor provide substantial novelty and additional elements to further understand how this signaling pathway works, from previous publications.

MAJOR:

a) A receptor-sugar structure would be ideal to talk about the perception of EPS (as mention in the manuscript). In the absence of that the authors, looking at the structure, could identify and mutate potential ligand motifs to identify key elements responsible for ligand binding. Binding of EPS to these mutants could be tested using either the BLI (previously used by the authors to describe the binding of EPS to the receptor) or the MST methods described. Gel filtrations and SDS gels of the mutants should be shown in the supplement to report the integrity of the mutants. To get receptor-sugar complexes is not trivial, however, different strategies can be tried. Despite the non-effect of the EPS truncated version in vivo (the penta-glycan), did the authors test binding to EPR3? If that would be the case, a shorter sugar could help in crystallization experiments. Also soaking pre-existing crystals with the ligand. That may not provide a full active complex but it would help to identify the motifs involved in binding in the receptor.

b) These structure ligand-perception mutants should be also be validated in vivo in complementation assays on the *epr3* mutant background.

c) The finding that similar EPS polymers also exists in other rhizobia is interesting. The authors already report the binding of EPS to EPR3 in a previous publication using a different method (Nature, 2015), however, they cross validate their previous results using a different binding method. In this manuscript the authors highlight the fact that EPR3 can also sense EPS polymers from other bacteria (*R. leguminosarum* and *S. meliloti*). The authors also mention that these bacteria are not really capable of infecting Lotus. How do the authors explain that phenomena with the binding data that they show in the manuscript? Specially in the case of *S. meliloti*, since it has a significant better Kd compared to the R7A EPS?

d) The authors also explored the role of the ligand O-acetyl groups in the binding to the receptor, and they find no apparent difference when compared to wt acetylated EPS. The authors should test the bioactivity of this deAOC-EPS ligand and see if these modifications are still relevant for the bioactivity of the ligand.

e) The observation of this protruding stem-like structure is very interesting, and the authors

should use this information to learn about the functioning of their receptor. The authors could try complementing in vivo a receptor variant lacking this region to see if indeed it is a critical feature for the receptor to signal. Controls of proper membrane localization of the receptor variant should be shown alongside.

Reviewer #2 (Remarks to the Author):

Wong et al describe the structure of a new class of carbohydrate binding receptor. In legumes EPR3 can perceive exo-polysaccharides (EPS) from rhizobial bacteria. This recognition has an important regulatory role in root nodule formation. This is a very well written and presented manuscript. Carbohydrate binding receptors play a crucial role in many host microbe interactions and I think the findings presented here would be of general interest to a wide range of scientists. In my opinion the results and outcomes seem appropriate for the target journal.

Wong and colleagues studied the ecto-domain of EPR3, which consist of three sub-domains M1, M2 and LysM3. They determined the crystal structure of the ecto-domain of EPR3. Presumably this was not a trivial undertaking as the authors utilised a nanobody approach to achieve this. Nanobodies are utilised in crystallisation studies to help stabilise the protein of interest and/or promote crystal contacts. Some discussion of why this approach was required for EPR3 within the manuscript would be welcomed.

The structure of EPR3 demonstrate that M1 has a novel fold associated with carbohydrate binding proteins. M2 also differs somewhat from other canonical LysM domains typically associated with carbohydrate binding. The authors use microscale thermophoresis to demonstrate that EPR3 can bind EPS. They also demonstrate promiscuity in EPS binding (ie EPS from non-compatible and compatible bacteria and de-O-acetylated EPS). Importantly, they show that EPR3 can distinguish carbohydrate ligands, ie no binding to chitohexose is observed. Does the modelling of the binding data assume a ratio (ie 1:1), and if so this should be mentioned and justified? To the best of my knowledge other Carbohydrate receptors have been shown to bind multiple ligands with differing affinities, could this be an additional distinguishing feature with EPR3-type receptors? The authors show that ligand binding is not impacted by the Nb186 (ED Fig 7), this is an important control but I could not see this mention within the manuscript.

The SAXS data for EPR3 is not in agreement with the structure suggesting the solution and crystal structures differ. This was improved significantly by modelling the stem region, which was not observed in the crystals structure, presumed because of flexible. While the assumption about the stem and its inclusion in the SAXS experiments might be probable, in my opinion there are still additional reasons/explanations for the deviation between the crystal structure and the SAXS data. For example, it is plausible that the M1, M2 and LySM3 domains themselves occupy alternative-extended conformations in solution. This may help explain why nanobodies were required for crystallisation, ie stabilisation of a single conformation. This could be validated further experimentally by performing SAXS analysis of a stem truncated EPR3 protein. In my opinion to conclude that it is the stem such an experiment would be necessary? That said, I don't think this experiment is required for publication but some acknowledgement of alternative reasons for the SAXS results should otherwise be included in text.

I also had some further questions regarding the stem. There are some comments in text that the stem could be involved as a spacer or be involved in interactions with co-receptors? Is this region of the protein conserved, in sequence and length between other plant homologues? this may give some further support to a conserved role/function of the stem and inclusion of the stem in the sequence alignment in ED Fig. 3 would be worth considering. Despite this, even if there is strong conservation in this region the inclusion of a putative co-receptor in Figure 3e seems premature and too speculative at this stage.

A big question that this study does not currently address is the ligand binding site. Undoubtedly this is something that the authors are currently pursuing and likely worthy of publication as a separate study. Despite this I was wondering if it would be possible, based on other known structures to include any prediction around ligand binding sites based on previous studies. Does the Nb186 (it does not appear to prevent ligand binding) exclude any binding sites shown for other LysM domains. Perhaps the compact nature of the paper restricts this but some comments concerning the ligand binding site(s) would be welcomed.

To conclude there are some important discussion points that I don't feel are currently addressed in the paper. For example, the authors present data showing that EPR3 is conserved across plants and this is used to help support the general importance and interest of the work. Yet it remains unclear to the reader the importance of these homologous receptors in different plant species. Some comments on previous studies involving the importance of homologues (if conducted) would be a welcome addition to the paper.

Minor comments.

The presentation of the figures in Fig 1 is great for comparison but the colour scheme makes it very difficult to distinguish M1 and M2, consider using more contrasting colours.

Indication of a consensus sequence would be useful in the alignments... Fig 2a and ED Fig 3

Figure 3e – The inclusion of a co-receptor is pure speculation and does not add anything to the work. Could be misleading to readers.

Reviewer #3 (Remarks to the Author):

The manuscript by Wong and colleagues, describes a detailed structural analysis of the extracellular domain of the presumed exopolysaccharide (EPS) receptor EPR3 from *Lotus japonicus*. They reveal a unique structure, deviating from homolog LysM-type receptor kinases, that is a structural signature for this new class of receptors in a wide range of plant species. Binding studies further show that EPR3 is a promiscuous receptor directly recognizing EPS.

The work is written and presented clearly and offers an important contribution by giving structural insight into a novel class of receptors in plants.

To strengthen the manuscript I have a few questions/suggestions:

The authors use a miniature antibody to purify the ectodomain of EPR3 produced in insect cells. Perhaps the authors can clarify or discuss whether the use of such an antibody may influence the 3D structure of the ectodomain. Is this structure also *ab initio* predicted based on modelling approaches (not using the obtained 3D structure as basis)? Did the authors also try different production systems to produce the EPR3 ectodomain?

It is mentioned that the b-a-bb fold structure of the M1 domain is conserved in EPR homologs from different plant species. Does this also hold for the M2 domain? And similar as the question before, does *ab initio* modelling of the M2 domain agree with the nanobody derived structural data?

The authors test the binding of the EPR3 ectodomain to CO6 and show that this EPR3 is unable to bind CO6. It would strengthen the manuscript if also the binding of the ectodomain to LCO's would be tested and modelled to rule out a potential binding of these molecules, which are relevant in the setting of rhizobial infections. Furthermore, to rule out any doubts about the specificity, the binding studies should also be performed with a different purification of the ectodomain (without

the nanobody), as done for example in the original 2015 paper.

Perhaps the authors can also model the binding characteristics of the mutated EPS molecules that are produced by to explain the observed phenotypes in their 2015 / 2017 papers. Is there a link with a difference in affinity for these different EPS molecules? Or is the discrimination based on the presumed co-receptors?

Reviewer #1 (Remarks to the Author):

Review: Structural signatures in EPR3 define a new class of plant carbohydrate receptors.

In their manuscript, Wong et al., determine and describe the apo structure of the *Lotus japonicus* EPR3 receptor ectodomain. The structure reveals a unique domain arrangement to sense carbohydrate ligands. The authors also explore some structural features of the receptor-ECD in solution using SAXs. SAXs data reveal an interesting protruding stem-like structure (not visible in the crystal structure) with unknown function. Using SAXs analysis of the receptor, in the presence or absence of the EPS ligand, they hypothesize that the receptor-ligand complex may require a potential co-receptor for activation. The authors also define the binding affinities of different EPS ligands to the receptor, showing its capacity to recognize different EPS ligands from different bacterial species, however, being able to distinguish other defense signalling sugars such as chitin. The authors also test in vitro the potential relevance of the O-acetyl groups for the recognition of the sugar ligand by the receptor.

The paper is nicely written and easy to follow, and the receptor ECD apo structure is indeed a novel element in the manuscript to be acknowledged and will provide a great platform to move forward in the understanding on how this signaling mechanism works; however, the authors do not exploit the structural data results nor provide substantial novelty and additional elements to further understand how this signaling pathway works, from previous publications.

Response: We are very pleased to learn that Reviewer 1 shares our enthusiasm and recognises the novelty that we hope will advance further studies in understanding signal transduction.

MAJOR:

a) A receptor-sugar structure would be ideal to talk about the perception of EPS (as mentioned in the manuscript). In the absence of that the authors, looking at the structure, could identify and mutate potential ligand motifs to identify key elements responsible for ligand binding. Binding of EPS to these mutants could be tested using either the BLI (previously used by the authors to describe the binding of EPS to the receptor) or the MST methods described. Gel filtrations and SDS gels of the mutants should be shown in the supplement to report the integrity of the mutants.

Response: We have indeed followed the mutation strategy suggested by reviewer 1. M1 is a completely novel fold, so there is no prior knowledge pointing to a putative ligand binding site. Nonetheless, we generated two EPR3 versions mutated in M1. We introduced a bulky amino acid in the flexible loop region following beta strand 1 and another version mutating a conserved hydrophobic amino acid in beta strand 3. Since the M2 domain associates with LysM domains and LysM3 is a conserved LysM domain, we could identify regions that corresponded to the chitin-binding groove in *Arabidopsis* CERK1 LysM2 domain (Liu et al. 2012, Science) by structural superposition. We introduced bulky amino acids in the putative ligand-binding regions in M2 and LysM3 respectively. Even though it is a costly and time-consuming procedure we managed to express and purify all mutants using our insect cell expression system and their SEC profiles resembled that of wild type proteins, which is indeed a good quality indicator for the integrity of the mutants, as pointed out by the reviewer. Using MST we tested the binding of these four mutants to EPS ligands, as well as chitin polymers, and did not detect any difference in affinities compared to wild-type protein. In addition, we tried to express and purify the P201L protein mutant in the LysM3 domain that previously led

to the identification of EPR3 in a suppressor screen (Kawaharada et al. 2015, Nature). The P201L mutant expressed poorly and it is very likely that the infection phenotype observed (the evasion of defective EPS surveillance in *Lotus japonicus*) is due to low levels of EPR3 (effectively a null mutant). We fully agree with the reviewer that identifying key residues/elements in the ligand binding site is a natural next step and we have so far not succeeded despite multiple attempts.

To get receptor-sugar complexes is not trivial, however, different strategies can be tried. Despite the non-effect of the EPS truncated version in vivo (the penta-glycan), did the authors test binding to EPR3? If that would be the case, a shorter sugar could help in crystallization experiments. Also soaking pre-existing crystals with the ligand. That may not provide a full active complex but it would help to identify the motifs involved in binding in the receptor.

Response: Truncated penta-glycan (ExoU) has an EPR3-dependent negative effect on infection thread formation (and subsequent mature nodule formation) *in planta*. Efforts to crystallise the ectodomain of EPR3 have been ongoing since 2014 with different carbohydrate ligands. We managed to obtain crystals only after we immunised a llama with EPR3 and selected the specific EPR3 nanobody Nb186. We were then able to solve the structure in the space group P 1 21 1 that show a very compact packing, which might unfortunately occlude the ligand binding site. Despite numerous co-crystallisation and soaking experiments (we have collected data and solved the structure from more than 30 crystals so far with different carbohydrate ligands included) we have not yet obtained the desired ligand-bound receptor complex.

b) These structure ligand-perception mutants should also be validated in vivo in complementation assays on the epr3 mutant background.

Response: As mentioned above, we have not identified any ligand-perception difference among the aa substitution variants tested that could justify *in vivo* experiments.

c) The finding that similar EPS polymers also exists in other rhizobia is interesting. The authors already report the binding of EPS to EPR3 in a previous publication using a different method (Nature, 2015), however, they cross validate their previous results using a different binding method. In this manuscript the authors highlight the fact that EPR3 can also sense EPS polymers from other bacteria (*R. leguminosarum* and *S. meliloti*). The authors also mention that these bacteria are not really capable of infecting Lotus. How do the authors explain that phenomena with the binding data that they show in the manuscript? Specially in the case of *S. meliloti*, since it has a significant better Kd compared to the R7A EPS?

Response: To clarify, EPR3 has higher affinity to *R. leguminosarum* and *M. loti* EPS compared to *S. meliloti* EPS. Our binding studies suggest that EPR3 is a promiscuous receptor capable of surveying EPS from different bacterial species while still selectively discriminating chitin and maybe other glycans from pathogens. This further suggests that EPS from different rhizobial species is perceived as more or less compatible even by legume species they do not normally infect. The observation that an *R. leguminosarum* DZL strain engineered to produce a Nod factor that triggers nodulation of Lotus can form infected and functional nodules supports this notion (Pacios Bras et al. 2000, MPMI and Radutoiu et al. 2007, EMBO J). This results has now been included and discussed in the manuscript, see line 144 - 156.

d) The authors also explored the role of the ligand O-acetyl groups in the binding to the receptor, and they find no apparent difference when compared to wt acetylated EPS. The au-

thors should test the bioactivity of this deAOC-EPS ligand and see if these modifications are still relevant for the bioactivity of the ligand.

Response: The phenotype of rhizobial exopolysaccharide mutants and the expression pattern of the Epr3 receptor suggest a major role for exopolysaccharide perception during the formation of root hair infection threads (Kawaharada et al. Nature 2015, Nature Comms 2017). Infection threads are only formed in the presence of rhizobia in a very limited number of root hair cells. In these responsive root hairs, an infection pocket/chamber of bacteria is formed by root tips curling up in a “shepherds crook” structure at the onset of infection thread development. At this stage, *Epr3* expression is upregulated in the responsive root hairs. We have so far been unable to establish a method for measuring bioactivity of purified or semi-purified glycan ligands in this system, most likely because externally applied ligands cannot access these few shielded infection sites in sufficient amounts to compete with the exopolysaccharide made by the rhizobia in the infection pocket or infection chamber. The role of deAOC-EPS ligands would have to be done by impairing the EPS specific acetylation (by inactivating several acetyl-transferase genes) using rhizobial mutants and at this point we have not identified such specific acetylation mutants in *M. loti*.

e) The observation of this protruding stem-like structure is very interesting, and the authors should use this information to learn about the functioning of their receptor. The authors could try complementing *in vivo* a receptor variant lacking this region to see if indeed it is a critical feature for the receptor to signal. Controls of proper membrane localization of the receptor variant should be shown alongside.

Response: Please see point above regarding *in vivo* complementation experiments. The discovery of EPR3 is relatively recent and our understanding of the mechanisms of EPS receptors is still limited. Future work focusing on the identification of co-receptors, potential role of the stem-like structure and other signalling components downstream of EPR3 are ongoing in our lab and would definitely guide *in vivo* studies suggested by reviewer 1 that are currently beyond the scope of this study.

Reviewer #2 (Remarks to the Author):

Wong et al describe the structure of a new class of carbohydrate binding receptor. In legumes EPR3 can perceive exo-polysaccharides (EPS) from rhizobial bacteria. This recognition has an important regulatory role in root nodule formation. This is a very well written and presented manuscript. Carbohydrate binding receptors play a crucial role in many host microbe interactions and I think the findings presented here would be of general interest to a wide range of scientists. In my opinion the results and outcomes seem appropriate for the target journal.

Response: We greatly appreciate these positive comments.

Wong and colleagues studied the ecto-domain of EPR3, which consist of three sub-domains M1, M2 and LysM3. They determined the crystal structure of the ecto-domain of EPR3. Presumably this was not a trivial undertaking as the authors utilised a nanobody approach to achieve this. Nanobodies are utilised in crystallisation studies to help stabilise the protein of interest and/or promote crystal contacts. Some discussion of why this approach was required for EPR3 within the manuscript would be welcomed.

Response: We have revised the text to better reflect the lengthy process of generating and characterizing nanobodies to facilitate crystallisation of the EPR3-Nb186 complex. See line 46-49.

The structure of EPR3 demonstrate that M1 has a novel fold associated with carbohydrate binding proteins. M2 also differs somewhat from other canonical LysM domains typically associated with carbohydrate binding. The authors use microscale thermophoresis to demonstrate that EPR3 can bind EPS. They also demonstrate promiscuity in EPS binding (ie EPS from non-compatible and compatible bacteria and de-O-acetylated EPS). Importantly, they show that EPR3 can distinguish carbohydrate ligands, ie no binding to chitohexose is observed. Does the modelling of the binding data assume a ratio (ie 1:1), and if so this should be mentioned and justified? To the best of my knowledge other Carbohydrate receptors have been shown to bind multiple ligands with differing affinities, could this be an additional distinguishing feature with EPR3-type receptors? The authors show that ligand binding is not impacted by the Nb186 (ED Fig 7), this is an important control but I could not see this mention within the manuscript.

Response: To our knowledge, plant LysM-type receptors have only been structurally and biochemically demonstrated to bind ligands in one LysM domain - the LysM2 domain of rice CEBiP (Liu et al. 2016, Structure) and *Arabidopsis* CERK1 (Liu et al. 2012, Science). In those studies, dimerization of LysM receptors in the presence of a long chitin polymer have been proposed to be important for immune signalling. ITC experiments show that rice CEBiP binds tetrachitin and octachitin with stoichiometries close to 1 (Liu et al. 2016, Structure). While we cannot attain stoichiometric information from MST, we know from previous BLI data that the ectodomain of EPR3 directly perceives *M. loti* EPS and here the binding curves were fitted and with a 1:1 binding model (Kawaharada et al. 2015, Nature). As correctly pointed out by the reviewer we measured if Nb186 impacts ligand binding and find that the isolated EPR3-Nb186 complex still binds EPS with similar affinity as the nanobody free receptor (Supplementary Fig. 7B). Additionally, we also enzymatically removed the glycosylation on EPR3 and show that these do not impact ligand binding. We have revised the text to more clearly describe these two important controls. See line 107-110.

The SAXS data for EPR3 is not in agreement with the structure suggesting the solution and crystal structures differ. This was improved significantly by modelling the stem region, which was not observed in the crystals structure, presumed because of flexible. While the assumption about the stem and its inclusion in the SAXS experiments might be probable, in my opinion there are still additional reasons/explanations for the deviation between the crystal structure and the SAXS data. For example, it is plausible that the M1, M2 and LysM3 domains themselves occupy alternative-extended conformations in solution. This may help explain why nanobodies were required for crystallisation, ie stabilisation of a single conformation. This could be validated further experimentally by performing SAXS analysis of a stem truncated EPR3 protein. In my opinion to conclude that it is the stem such an experiment would be necessary? That said, I don't think this experiment is required for publication but some acknowledgement of alternative reasons for the SAXS results should otherwise be included in text.

Response: The reorganisation of M1, M2 and LysM3 domains are unlikely due to the fact that the three domains are held together in a compact structure stabilised by three central disulphide bridges. We directly observed this high stability of EPR3 when measuring the melting temperature of the receptor ectodomain to be 63.8 degrees Celsius. Following the good

advice of the reviewer we have computed a 50 ns molecular dynamics (MD) simulation and find that the overall fold is stable and maintained. We observe a few flexible loops in the MD simulation that are close to the nanobody binding epitope, which might indeed be stabilized by the nanobody and facilitate crystallization. Additionally, the flexible N-terminus of EPR3 is in proximity to all three complementarity-determining regions and is also stabilized by the nanobody. We have revised the text to reflect that the stem-like structure still needs to be validated *in planta* and understood functionally. See line 93-100.

I also had some further questions regarding the stem. There are some comments in text that the stem could be involved as a spacer or be involved in interactions with co-receptors? Is this region of the protein conserved, in sequence and length between other plant homologues? this may give some further support to a conserved role/function of the stem and inclusion of the stem in the sequence alignment in ED Fig. 3 would be worth considering. Despite this, even if there is strong conservation in this region the inclusion of a putative co-receptor in Figure 3e seems premature and too speculative at this stage.

Response: We have now included the sequence of the EPR3 stem in Fig. 3 together with the conservation Logo derived from sequence alignment of EPR3 homologues. This analysis shows that some conservation exists both in terms of amino acid composition (dominated by conserved glycine and lysine residues) and length. We have revised the text to better reflect that our speculative model with an as yet unidentified co-receptor is based not only on our SAXS data that show no ligand induced EPR3 homodimers, but relies on characterised single pass receptor systems where two receptors, typically with an active kinase (like EPR3), interact with a receptor with a pseudokinase co-receptor during signal transduction (Hayafune et al. 2014, PNAS and Cao et al. 2014, eLife). We have edited our model in figure 3f accordingly. See line 161-165.

A big question that this study does not currently address is the ligand binding site. Undoubtedly this is something that the authors are currently pursuing and likely worthy of publication as a separate study. Despite this I was wondering if it would be possible, based on other known structures to include any prediction around ligand binding sites based on previous studies. Does the Nb186 (it does not appear to prevent ligand binding) exclude any binding sites shown for other LysM domains. Perhaps the compact nature of the paper restricts this but some comments concerning the ligand binding site(s) would be welcomed.

Response: We agree that identifying and engineering the ligand binding site is an obvious next step. Please see our response to point (a) of reviewer 1 above regarding the challenges involved in identifying the ligand binding site. As for the nanobody bound receptor: Nb186 interacts with the EPR3 receptor in such a way that it does not occlude either M1, M2 or LysM3 completely. From our ligand binding studies of EPR3, we know that the nanobody does not enhance or inhibit ligand binding. As the reviewer correctly points to it would have been interesting if Nb186 blocked or inhibited ligand binding as this would give some insights into this new ligand binding site.

To conclude there are some important discussion points that I don't feel are currently addressed in the paper. For example, the authors present data showing that EPR3 is conserved across plants and this is used to help support the general importance and interest of the work. Yet it remains unclear to the reader the importance of these homologous receptors in different plant species. Some comments on previous studies involving the importance of homologues (if conducted) would be a welcome addition to the paper.

Response: EPS is involved in non-self recognition and responses to microbial associated molecular patterns. However, to our knowledge, EPR3 in *Lotus japonicus* is the first exopolysaccharide receptor identified in eukaryotes and our study points to a broader role of this new class of receptors in the plant kingdom. We agree that future studies in many different plant species will be very interesting to follow up on but this is only the first study demonstrating that this unique receptor family exists and is widely distributed in plants, so the publications in this area are simply lacking at the moment. See line 82-84 for changes in the manuscript highlighting this point.

Minor comments.

The presentation of the figures in Fig 1 is great for comparison but the colour scheme makes it very difficult to distinguish M1 and M2, consider using more contrasting colours.

Response: We changed the colour of the beta-sheets in Fig. 1 for better contrast when comparing M1, M2 and LysM3.

Indication of a consensus sequence would be useful in the alignments... Fig 2a and ED Fig 3

Response: We added the consensus sequence to Fig. 2a and supplementary Fig. 3.

Figure 3e – The inclusion of a co-receptor is pure speculation and does not add anything to the work. Could be misleading to readers.

Response: We agree that our model is very speculative and we have revised the text to better reflect that our model with an as yet unidentified co-receptor is based not only on our SAXS data that shows no ligand-induced homodimers, but also on characterised single pass receptor systems where two receptors, typically with an active kinase (like EPR3), interact with a receptor with a pseudokinase co-receptor during signal transduction. See line 161-165.

Reviewer #3 (Remarks to the Author):

The manuscript by Wong and colleagues, describes a detailed structural analysis of the extracellular domain of the presumed exopolysaccharide (EPS) receptor EPR3 from *Lotus japonicus*. They reveal a unique structure, deviating from homolog LysM-type receptor kinases, that is a structural signature for this new class of receptors in a wide range of plant species. Binding studies further show that EPR3 is a promiscuous receptor directly recognizing EPS. The work is written and presented clearly and offers an important contribution by giving structural insight into a novel class of receptors in plants.

To strengthen the manuscript I have a few questions/suggestions:

Response: We thank the reviewer for these positive comments and for sharing our excitement of this novel class of plant receptor.

The authors use a miniature antibody to purify the ectodomain of EPR3 produced in insect cells. Perhaps the authors can clarify or discuss whether the use of such an antibody may influence the 3D structure of the ectodomain. Is this structure also *ab initio* predicted based on modelling approaches (not using the obtained 3D structure as basis)? Did the authors also try different production systems to produce the EPR3 ectodomain?

Response: We purify the ectodomain of EPR3 expressed in insect cells and do not as such use nanobodies for the purification of the protein. As EPR3 alone did not yield crystals, we turned to nanobodies as in many studies these have proven to be excellent crystallisation chaperons. Indeed, using this approach we managed to obtain crystals that led to the successful structure determination of the EPR3-Nb186 complex. We show in the manuscript that ligand binding is not affected by the nanobody, which is a good indicator that the nanobody binding does not disturb the structure of EPR3. Additionally we have now run a molecular dynamics (MD) simulation of EPR3 and the EPR3-Nb186 complex and see that the structures are very stable and that no structural changes were induced by Nb186. We have over the years experimented with many different expression systems (bacteria, yeast, plant etc.) for these types of receptor ectodomains. So far, the insect cell expression system is the preferred system that very importantly allows for correct folding and formation of the three conserved disulfide bridges in EPR3 and where we can produce quantities sufficient for biochemical and structural characterisation.

It is mentioned that the b-a-bb fold structure of the M1 domain is conserved in EPR homologs from different plant species. Does this also hold for the M2 domain? And similar as the question before, does *ab initio* modelling of the M2 domain agree with the nanobody derived structural data?

Response: This is an excellent question. Our focus has mainly been on the novelties in the M1 domain topology/folding and here 15 independent *ab initio* force-field based models of EPR3 homologues show this new feature. Our rationale was primed by the observation that M1 is a new fold in carbohydrate binding proteins and M2 has homology to the known LysM domain. We have now modelled M2 and indeed the folding and topology of the M2 domain agrees with the crystal structure, showing that non-homology force-field based modelling is a good approach for predicting the folds of small domains. There is however a size limitation to what this modelling program can reliably fold, and inclusion of a nanobody and full ectodomains is currently not feasible. As mentioned before, we validated our EPR3 nanobody complex structure by showing in the manuscript that ligand binding (and therefore structure) is not affected by Nb186 binding. See line 107-110.

The authors test the binding of the EPR3 ectodomain to CO6 and show that this EPR3 is unable to bind CO6. It would strengthen the manuscript if also the binding of the ectodomain to LCO's would be tested and modelled to rule out a potential binding of these molecules, which are relevant in the setting of rhizobial infections. Furthermore, to rule out any doubts about the specificity, the binding studies should also be performed with a different purification of the ectodomain (without the nanobody), as done for example in the original 2015 paper.

Response: Following the procedures and conditions used to detect EPS binding to EPR3 in BLI binding experiments (Kawaharada et al. 2015, Nature), and despite showing robust *M. loti* Nod factor binding to *Lotus* Nod factor receptors using this approach (Murakami et al. 2018, eLife), we have so far not detected *M. loti* Nod factor binding and have now revised the text to include this. See line 119 – 124.

Perhaps the authors can also model the binding characteristics of the mutated EPS molecules that are produced by to explain the observed phenotypes in their 2015 / 2017 papers. Is there

a link with a difference in affinity for these different EPS molecules? Or is the discrimination based on the presumed co-receptors?

Response:

Different affinities to different carbohydrate ligands is one level of discriminating ligands and we have so far been unable to convincingly model ligand binding. Signalling by LysM receptors typically involve additional co-receptors that could provide an additional level of ligand discrimination and modulate signalling output. The current model for receptor signalling relies on receptor complex formation with at least one LysM receptor with an active kinase and a co-receptor with a pseudokinase or no kinase domain. To state a few examples, NFR1 cannot promote symbiosis signalling without its NFR5 counterpart (Radutoiu et al. 2007, EMBO J and Ried et al. 2014, eLife) and LYK5 is required for CERK1 chitin-triggered immunity (Cao et al. 2014, eLife). Studies have also shown that intermolecular dimerization can occur on a long ligand (Hayafune et al. 2014, PNAS, Liu et al. 2012, Science). Our SAXS studies show that the ectodomain of EPR3 remains a monomer when bound to EPS, we therefore very cautiously suggest that EPR3 might recruit an as yet uncharacterised co-receptor for activation and signalling output and this might very well be different with the different EPS ligands as pointed out by the reviewer. See line 161-165.

REVIEWERS' COMMENTS:

Reviewer #2 (Remarks to the Author):

The authors have provided a comprehensive and measured rebuttal and subsequent revision of the paper. In doing so I feel they have addressed the comments and concerns that I raised from my initial review of the work.

Your Sincerely,
Simon Williams

Reviewer #3 (Remarks to the Author):

The revised manuscript by Wong and co-workers sufficiently addressed previous concerns/comments. The clearly written and presented work offers a first structural basis for this novel type of carbohydrate-binding receptor, binding EPS, that may play important roles in a wide range of plant-microbe interactions, in addition to its important role in the rhizobium-legume symbiosis.

REVIEWERS' COMMENTS:

Reviewer #2 (Remarks to the Author): The authors have provided a comprehensive and measured rebuttal and subsequent revision of the paper. In doing so I feel they have addressed the comments and concerns that I raised from my initial review of the work.

Your Sincerely,
Simon Williams

Response: We thank Simon Williams for taking the time to review our paper, and in the process, helped us to improve the quality of the paper. We are pleased that we have been able to sufficiently address his comments and concerns.

Reviewer #3 (Remarks to the Author):

The revised manuscript by Wong and co-workers sufficiently addressed previous concerns/comments. The clearly written and presented work offers a first structural basis for this novel type of carbohydrate-binding receptor, binding EPS, that may play important roles in a wide range of plant microbe interactions, in addition to its important role in the rhizobium-legume symbiosis.

Response: We thank Reviewer #3 for taking the time to review our paper, and in the process, helped us to improve the quality of the paper. We are pleased that we have been able to sufficiently address his comments and concerns.